Journal of Machine Learning Research 30 (2025) 1-12   Submitted 30/06/25; Revised 08/08/25; Published xx/xx/xx

# Context-guided Prompt Learning
# for Continual WSI Classification

**Giulia Corso**[*]                                        GIULIA.CORSO@UNIMORE.IT
**Francesca Miccolis**[*]                          FRANCESCA.MICCOLIS@UNIMORE.IT
**Angelo Porrello**                                ANGELO.PORRELLO@UNIMORE.IT
**Federico Bolelli** ✉                             FEDERICO.BOLELLI@UNIMORE.IT
**Simone Calderara**                             SIMONE.CALDERARA@UNIMORE.IT
**Elisa Ficarra**                                       ELISA.FICARRA@UNIMORE.IT
*University of Modena and Reggio Emilia, Italy*

## Abstract

Whole Slide Images (WSIs) are crucial in histological diagnostics, providing high-resolution insights into cellular structures. In addition to challenges like the gigapixel scale of WSIs and the lack of pixel-level annotations, privacy restrictions further complicate their analysis. For instance, in a hospital network, different facilities need to collaborate on WSI analysis without the possibility of sharing sensitive patient data. A more practical and secure approach involves sharing models capable of continual adaptation to new data. However, without proper measures, catastrophic forgetting can occur. Traditional continual learning techniques rely on storing previous data, which violates privacy restrictions. To address this issue, this paper introduces Context Optimization Multiple Instance Learning (CooMIL), a rehearsal-free continual learning framework explicitly designed for WSI analysis. It employs a WSI-specific prompt learning procedure to adapt classification models across tasks, efficiently preventing catastrophic forgetting. Evaluated on four public WSI datasets from TCGA projects, our model significantly outperforms state-of-the-art methods within the WSI-based continual learning framework. The source code is available at `https://github.com/FrancescaMiccolis/CooMIL`.

**Keywords:**   WSI, Multi-instance Learning, Prompt Learning, Continual Learning

## 1 Introduction

Whole Slide Images (WSIs) are valuable tools in digital pathology and clinical diagnostics (Bontempo et al., 2023a). In addition to the vast dimensions and lack of precise pixel-level annotations (Lu et al., 2020; Huang et al., 2022; Javed et al., 2020; Ponzio et al., 2020)—WSIs are often subject to privacy restrictions, which hinder data sharing and collaboration (Bisson et al., 2023; Kanwal et al., 2023; Bandeira et al., 2023). Continual Learning (CL) offers a solution by allowing models to learn incrementally from data distributed across multiple healthcare institutions, while respecting patient privacy and data governance policies. Indeed, fine-tuning even large-scale models on new datasets often leads to *catastrophic forgetting*, where the model forgets previously learned information (Robins, 1995). This issue is critical in WSI analysis, where tissue subtypes and treatment protocols evolve rapidly. CL aims to mitigate this problem and enhance model adaptability to new datasets and tasks.

---

[*]Equal contribution. Authors are allowed to list their name first on their CVs.

Among various CL methodologies such as regularization (Aljundi et al., 2018; Zenke et al., 2017) and architectural strategies (Rusu et al., 2016), only rehearsal-based models seem to be effective against catastrophic forgetting in WSI analysis (Huang et al., 2023a). However, such approaches rely on memory buffers, which are impractical in privacy-sensitive medical contexts. To address these challenges, we propose a rehearsal-free strategy for WSI analysis, employing a multimodal multi-resolution classifier with an attribute word bank that pairs unique identifiers (keys) with prompts enriched with context-derived information. The classifier is also based on Multiple Instance Learning (MIL) (Carbonneau et al., 2018; Panariello et al., 2022), which is applied to WSI classification (Li et al., 2021a; Bontempo et al., 2023b), where each bag of patches is annotated at the slide level.

***Contributions.*** In addressing the outlined challenges and introducing a novel architecture for the analysis of WSIs within a continual learning framework, our work makes several significant contributions to the field of medical image analysis: *(i)* the proposed approach overcomes the limitations of traditional continual learning strategies on WSIs, which rely on rehearsing previous data to prevent catastrophic forgetting. Our method offers a more efficient and privacy-compliant solution for continual learning in WSI analysis. *(ii)* We develop a novel prompt-learning-based MIL in WSI analysis. Different from other strategies, it exploits a MIL approach to contextualize prompts. *(iii)* Additionally, we introduce a novel solution of prompt learning tailored to the multi-resolution characteristics of WSIs, enabling our model to focus effectively on relevant features across different scales.

## 2 Related Work

***Continual Learning (CL) for Histology.*** Continual learning—the ability to incrementally acquire new knowledge while retaining previously learned information—is vital in medical image analysis. It is generally categorized into three main approaches: *(i)* regularization-based methods, which penalize changes using various regularization terms (Kirkpatrick et al., 2017; Li and Hoiem, 2017; Zenke et al., 2017); *(ii)* rehearsal-based strategies, which retain and replay past data during training (Li and Hoiem, 2017; Chaudhry et al., 2019; Caccia et al., 2022); and *(iii)* architectural solutions, which expand the model's parameters to accommodate new tasks (Rusu et al., 2016). Notably, some works have investigated continual learning in pathology (Derakhshani et al., 2022; Veena et al., 2022; Thandiackal et al., 2024). However, these efforts are primarily limited to patch-level analysis. Addressing these challenges at the slide level, ConSlide (Huang et al., 2023a) introduces a continual learning framework specifically designed for WSI classification. ConSlide employs a rehearsal-based strategy by maintaining a memory buffer of representative slide-level features from previous tasks. During training, it replays this stored data to mitigate catastrophic forgetting, effectively balancing the learning of new and old tasks. While ConSlide advances continual learning in digital pathology, its reliance on rehearsal methods introduces concerns related to data storage and potential privacy issues, critical factors in medical applications where data security is crucial. In contrast, our work is the first to propose a rehearsal-free continual learning framework for WSI classification.

***Continual Prompt Learning.*** Vision-Language Models (Jia et al., 2021; Radford et al., 2021) showcased remarkable capabilities in learning versatile visual representations on standard benchmarks (Radford et al., 2021) and histological data (Huang et al., 2023b). Building

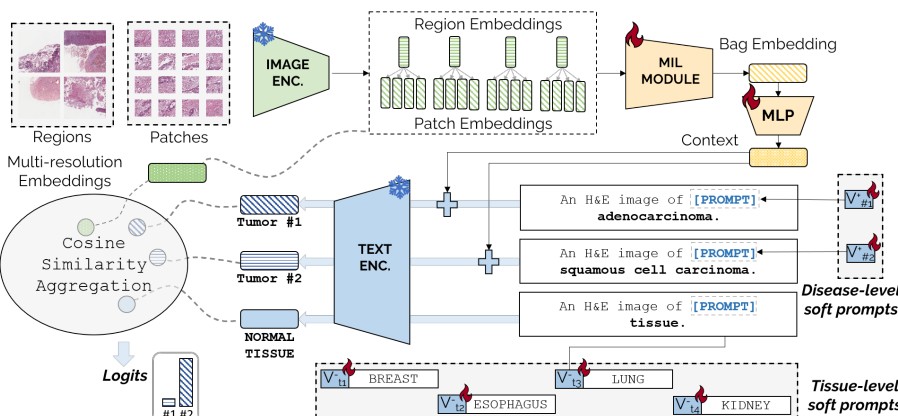

Figure 1: WSIs are decomposed into regions and patches, from which multi-resolution embeddings are extracted using a frozen image encoder (top left). These features are aggregated using a MIL module (top right) to produce a context-aware bag embedding. The context is injected into learnable soft prompts which are processed by a frozen text encoder (center). Classification logits at the slide level are obtained by computing the cosine similarity between the visual and textual embeddings (bottom left).

on these advancements, recent research has focused on optimizing training methodologies for specific classification tasks, moving beyond conventional model fine-tuning to prevent a degradation of the representation space (Gao et al., 2023; Wortsman et al., 2022; Yao et al., 2024; Zhang et al., 2021; Dong et al., 2019; He et al., 2016). In particular, recent efforts in continual learning have introduced visual prompt tuning (Wang et al., 2022a,b), integrating a minimal set of adaptable parameters directly into the input, thereby furnishing the pre-trained models with additional guidance for enhanced performance on downstream tasks (Li et al., 2021b). L2P (Wang et al., 2022b) bridges visual prompting with continual learning, utilizing a shared prompt pool for task sequence adaptation. In this regard, recent works (Frascaroli et al., 2024) apply prompt-tuning to multi-modal architectures in order to learn a sequence of classification tasks without forgetting. Recent works (Ranem et al., 2024) explore continual learning in medical imaging, focusing on volumetric data. However, none of the aforementioned prompting strategies can be directly applied to the gigapixel nature of WSIs.

## 3 Method

***Overview.*** To address the challenges of continual WSI classification preserving privacy, we propose a solution that integrates rehearsal-free continual learning techniques with multi-instance learning. The proposed model comprises several components designed to enable effective classification in a continual learning setting. The pipeline (Fig. 1) starts with an image encoder that fuses features extracted from patches at multiple resolutions (Sec. 3.1). Subsequently, a context-aware MIL module provides a bag-level representation, which is injected into learnable soft prompts that are then processed by a text encoder (Sec. 3.2). Classification logits are obtained by computing cosine similarities between the visual and textual embeddings (Sec. 3.3). In addition, a continual word bank (Fig. 2) facilitates the dynamic retrieval of the most relevant prompts over time, as detailed in Sec. 3.4.

### 3.1 Multi-scale Slide Representation

We compute a multi-scale representation for each WSI employing the image encoder $f(\cdot)$ of a foundation model pre-trained on histological images (Lu et al., 2024). Each slide is represented as a multi-resolution embedding $B$ (green box in Fig. 1) of instances $x_i$:

$$B = \{x_1, ..., x_n\}; \quad x_i = \bigcup_{\forall p_j \in r_i;} \{\frac{f(p_j) + f(r_i)}{2}\}; \tag{1}$$

where $r_i$ denotes the $i$-th region at a coarser resolution, and $p_j$ are the patches at a finer resolution within $r_i$. The notation $p_j \in r_i$ indicates that patch $p_j$ is contained within region $r_i$. Averaging features from finer patches $p_j$ and their corresponding coarser regions $r_i$ captures both macro and micro details, yielding robust representations.

### 3.2 Context-aware Prompt Learning

Textual prompts often generalize poorly (Zhou et al., 2022). To address this, we draw inspiration from DSMIL (Li et al., 2021a) and CoCoOp (Zhou et al., 2022), and propose a mechanism to inject image-derived contextual information directly into the prompts, enhancing their relevance and adaptability.

Specifically, in this MIL module, the instance-level representation $x_i$ is transformed into two vectors, corresponding to query $q_i$ and value $v_i$, computed as: $q_i = W_q x_i$, $v_i = W_v x_i$ with $i = 0, \ldots, N-1$, where $W_q$ and $W_v$ are learnable weight matrices. We use a distance measurement $U$ to quantify the similarity between an arbitrary instance and the critical instance $x_{crit}$, selected using max pooling, as in Eq. (2):

$$U(x_i, x_{crit}) = \frac{\exp(\langle q_i, q_{crit} \rangle)}{\sum_{k=1}^{n} \exp(\langle q_k, q_{crit} \rangle)}; \quad (2) \qquad b = \sum_{i=1}^{n} U(x_i, x_{crit}) \, v_i. \tag{3}$$

For constructing the bag embedding $b$ in Eq. (3), we perform an element-wise weighted sum of the value vector $v_i$ across all instances using $U(x_i, x_{crit})$ as weights. To facilitate the information flow from the MIL module to the learnable prompts, we employ a MLP denoted as $M$. The context to be injected in the prompt, denoted as $\pi$ (yellow box in Fig. 1) is obtained as $\pi = M(b)$ (4).

In the proposed model, we introduce two distinct prompts with complementary functions, i.e., localization and classification. Each prompt is composed of a static template ("An H&E image of"), a learnable word embedding vector $V$, and a given class name CLS:

$$P^{tumor,normal} = \left[ TEMPLATE, V^{+,-}, \text{CLS}^{tumor,normal} \right]. \tag{5}$$

The $P^{normal}$ is designed to distinguish normal tissues from pathological anomalies, thus localizing areas of interest. The $P^{tumor}$, on the other hand, is used to classify the identified instances based on the type of tumor. This two-step process ensures that the system not only detects areas of concern but also provides a precise classification. Having defined both the context $\pi$, Eq. (4), and the base prompt $P^{tumor,normal}$, Eq. (5), the final prompt after context injection is obtained as $P_\pi^{tumor,normal} = P^{tumor,normal} + \pi$ and processed by the text encoder $e(\cdot)$.

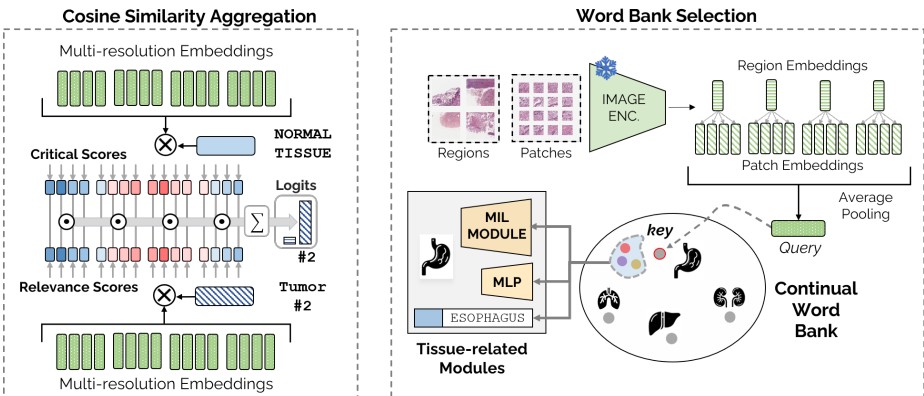

Figure 2: Cosine Similarity Aggregation (left) and Continual Word Bank (right). On the left, multi-resolution embeddings are matched against specific tumor and normal textual prompts. The comparison with normal tissue prompts allows for modulating tumor scores for the final prediction. On the right, a query representing the average instance embedding is compared with keys to select the corresponding set of parameters.

### 3.3 Cosine Similarity Aggregation

For each multi-scale instance feature $x_i$, Eq. (1), class $c$ and tissue $t$, we compute the cosine similarity between the visual and textual features as $S_{x_i,c}^{tumor} = \langle x_i, e(P_c^{tumor}) \rangle$ and $S_{x_i,t}^{normal} = \langle x_i, e(P_t^{normal}) \rangle$. To make a single prediction $S_c$ for the whole image, we aggregate the *relevance scores* $S_{x_i,c}^{tumor}$ modulated by the corresponding magnitude of the *critical scores* $S_{x_i,t}^{normal}$ (Fig. 2, left):

$$S_c = \sum_{x_i} \mathbb{1}_{c \in t} \left( exp(-S_{x_i,t}^{normal}) \cdot S_{x_i,c}^{tumor} \right). \tag{6}$$

In a nutshell, we compute the aggregated slide-level representation by summing the similarity score of the tumor regions weighted by the inverse of their degree of "normality".

### 3.4 Continual Word Bank

To work effectively within a continual learning framework, we introduce the Continual Word Bank (Fig. 2, right). This dynamic repository accumulates and refines task-specific prompts over time, each tailored to a tissue type and its associated classes. By selectively retrieving these prompts, the model can process a sequence of tasks without catastrophic forgetting. At each task $t_i$, we compute the average patch representation for each bag $B_h$ as in Eq. (7); then we apply the K-Means clustering algorithm to partition them into $k$ clusters. Each cluster $G_j$ is associated with a centroid $g_j$, Eq. (8), representing the mean feature vector of the points belonging to that cluster:

$$\bar{b}_h = \frac{1}{|B_h|} \sum_{x_i \in B_h} x_i; \qquad (7) \qquad\qquad g_j = \frac{1}{|G_j|} \sum_{\bar{b}_h \in G_j} \bar{b}_h. \qquad (8)$$

For each new task, centroids are computed on-the-fly and used as transient keys to select the most relevant prompts. In this key-value system, the stored values are the prompts and the MIL modules, which are lightweight components that ensure scalability and efficiency across tasks, without retaining identifiable patient information. During inference, the average instance representation for each new slide is computed as it was during training, and it

Table 1: Comparison of continual learning methods across different dataset orders.

| CL Type | Method | (a) $NSCLC \to BRCA \to RCC \to ESCA$ | | | (b) $ESCA \to RCC \to BRCA \to NSCLC$ | | |
|---|---|---|---|---|---|---|---|
| | | ACC (↑) | Task-ACC (↑) | Fgt.(↓) | ACC (↑) | Task-ACC (↑) | Fgt.(↓) |
| | Joint (Upper) | 91.6 ± 2.4 | 91.5 ± 3.2 | | 91.6 ± 2.4 | 91.5 ± 3.2 | |
| | Naïve (Lower) | 21.7 ± 3.0 | 38.4 ± 9.2 | 51.0 ± 12.9 | 32.6 ± 1.3 | 38.1 ± 5.6 | 75.7 ± 7.4 |
| Regularization-based | LwF | 19.7 ± 4.3 | 28.9 ± 0.9 | 44.2 ± 5.3 | 32.9 ± 1.6 | 39.8 ± 11.7 | 81.7 ± 15.6 |
| | EWC | 28.1 ± 2.6 | 56.0 ± 1.4 | 64.5 ± 4.1 | 46.6 ± 5.0 | 55.1 ± 1.8 | 77.9 ± 6.1 |
| Rehearsal-based | GDumb | 48.4 ± 12.2 | 18.1 ± 3.6 | 1.1 ± 2.3 | 42.8 ± 15.1 | 17.6 ± 7.7 | 6.0 ± 4.9 |
| | ER-ACE | 86.8 ± 2.9 | 87.8 ± 1.8 | 2.8 ± 1.4 | 88.8 ± 2.1 | 90.6 ± 2.7 | 7.3 ± 5.0 |
| | DER++ | 88.4 ± 1.2 | 90.3 ± 1.2 | 3.7 ± 0.7 | 89.6 ± 1.1 | 91.2 ± 3.1 | 5.6 ± 4.3 |
| | DER++ w/o buf. | 29.9 ± 3.8 | 57.9 ± 2.1 | 62.9 ± 5.2 | 48.6 ± 4.3 | 58.0 ± 2.1 | 79.2 ± 4.1 |
| | ConSlide | 66.3 ± 3.7 | 80.3 ± 1.5 | 25.8 ± 3.4 | 69.0 ± 3.8 | 81.8 ± 2.7 | 49.2 ± 5.4 |
| | ConSlide w/o buf. | 26.5 ± 4.2 | 54.8 ± 1.9 | 66.5 ± 5.5 | 37.6 ± 4.5 | 53.1 ± 2.6 | 89.2 ± 3.7 |
| Prompt-based | **CooMIL (Ours)** | **88.6 ± 2.7** | **90.7 ± 2.5** | **3.6 ± 1.4** | **89.9 ± 3.4** | **91.3 ± 2.4** | **5.1 ± 3.8** |

serves as a query to retrieve the most pertinent entries from the continual word bank via a nearest-neighbor search. During training, we minimize the Cross-Entropy loss:

$$L(B, y, t_i) = \mathop{\mathbb{E}}_{\forall (B,y) \in t_i} (y \cdot log(softmax(y_c))), \tag{9}$$

where $y_c$ represents the predicted class scores. For the text prompts, $y_c = S_c$, while for the MIL module, $y_c = W_{cls} \cdot b$, where $W_{cls}$ is a learnable weight and $b$ is defined in Eq. (3). Finally, the loss is computed as $L = L_{MIL} + L_{text}$. In a continual context, a set of parameters (including $V^+$, $V^-$, and the MIL module) is instantiated for each task $t_i$. Only parameters corresponding to the current task are optimized during training.

## 4 Experiments

### 4.1 Datasets

**Continual WSI benchmark.** To validate our proposed architecture in a class-incremental learning setting, we conducted experiments using an improved version of the benchmark introduced by ConSlide (Huang et al., 2023a). Class-incremental learning requires models to recognize new classes without forgetting previously learned ones. In this context, the order of the datasets plays a crucial role, particularly with datasets of varying sizes and complexities. Unlike the original ConSlide benchmark, our version explicitly accounts for dimensionality differences, class imbalance, and presentation order. Tab. 1 reports the mean and standard deviation of a 10-fold cross-validation performed on two task orders: one from the most to the least numerous (Tab. 1a), and its reverse (Tab. 1b). The data include four tumor types, each defining separate binary subtype classification tasks—**NSCLC**, **BRCA**, **RCC**, **ESCA**.

***Preprocessing.*** Each slide is processed with CLAM (Lu et al., 2021) to extract non-overlapping regions $r$ of dimensions $4\,096 \times 4\,096$ sampled at a resolution of up to 0.5 $\mu m/pixel$. Tiles are partitioned into 64 non-overlapping $512 \times 512$ patches $p$, resized to $224 \times 224$, and encoded with CONCH's vision encoder (Lu et al., 2024).

***Experimental Setting.*** Our model employs CooMIL as the backbone, while all continual learning baselines adopt HIT, the architecture from ConSlide (Huang et al., 2023a). To ensure fairness, all methods rely on the same CONCH-based embeddings. Models are

Table 2: (a) Task accuracy with varying numbers of centroids. (b) Impact of context and multi-scale on classification performance. (c) Impact of context type, here **V** denotes injecting context only into the learnable part of the prompt, and **P** refers to appending the context to the entire prompt. **Tumor** and **Normal** indicate where injection is performed.

| (a) Clusters per Task | | (b) Context and Multi-scale | | | (c) Context Type | | | |
| --- | --- | --- | --- | --- | --- | --- | --- | --- |
| # Centroids | Task-ACC | Context | Multi-scale | ACC | V/P | Tumor | Normal | ACC |
| 1 | $85.7 \pm 2.3$ | | | | V | ✓ | ✓ | $87.2 \pm 3.2$ |
| 4 | $86.5 \pm 1.0$ | ✗ | ✗ | $84.6 \pm 2.8$ | V | ✓ | ✗ | $85.2 \pm 3.1$ |
| 8 | $87.5 \pm 1.9$ | ✗ | ✓ | $86.6 \pm 2.0$ | V | ✗ | ✓ | $84.2 \pm 3.6$ |
| 12 | $87.2 \pm 2.0$ | ✓ | ✗ | $86.9 \pm 3.0$ | P | ✓ | ✓ | $\mathbf{88.6 \pm 2.7}$ |
| 14 | $\mathbf{90.7 \pm 2.5}$ | ✓ | ✓ | $\mathbf{88.6 \pm 2.7}$ | P | ✓ | ✗ | $86.2 \pm 2.2$ |
| 16 | $88.5 \pm 2.4$ | | | | P | ✗ | ✓ | $85.0 \pm 2.9$ |

trained for 50 epochs using the Adam optimizer with a learning rate of 0.0003, employing a 10-fold cross-validation approach.

**Evaluation Metrics.** Besides Accuracy (**ACC**), we report CL metrics (De Lange et al., 2021) such as Task Accuracy (**Task-ACC**), i.e., performance on the current task only, and Forgetting (**Fgt.**), the decline in accuracy on earlier tasks (Boschini et al., 2022). Metrics reported in Tab. 1 were saved at the end of the final task in a ten-fold validation fashion.

## 4.2 Experimental Results

**Continual Comparison** We evaluated our proposed model against several leading continual learning baselines, covering regularization and rehearsal approaches in Tab. 1. Regularization-based methods, such as LwF (Li and Hoiem, 2017) and EWC (Kirkpatrick et al., 2017), aim to mitigate forgetting by constraining updates to important parameters for previously learned tasks. On the other hand, rehearsal-based methods, including GDumb (Prabhu et al., 2020), ER-ACE (Caccia et al., 2022), DER++ (Buzzega et al., 2020), and ConSlide (Huang et al., 2023a), rely on storing and replaying a subset of past data to help retain knowledge. These models were evaluated with a fixed buffer size of 5 WSIs. Specifically, our model boosts overall accuracy by over 20% and task-specific accuracy by 10% compared to ConSlide, in both normal and reverse task orders. Although ConSlide relies on a memory buffer, it still suffers a high forgetting rate (25.8% in normal order, nearly doubling in reverse). By contrast, our model—without any buffer—achieves substantially lower forgetting rates of just 3.6% in normal order and 5.1% in reverse order. The only methods with metrics comparable to ours in both the order settings are ER-ACE and DER++. However, they rely on a memory buffer and require significantly higher computational resources than our model. These results underscore the superior performance of our approach, which avoids storing past data while still achieving better overall metrics.

## 4.3 Further Analysis

**What is the optimal number of centroids?** In Tab. 2(a), we explore how the number of centroids per task influences the performance of the task predictor. A large number of clusters can better represent the entire task variability. If too high, the centroids overfit

the training representation. A good balance is obtained considering 14 centroids per task, which achieves 90.7% in task identification accuracy.

***Are multi-scale and context effective?*** In Tab. 2(b), multi-scale representations consistently enhance performance. Without context, incorporating multi-scale features leads to a 2% increase in accuracy. When context is included, the addition of multi-scale representations yields a further 1.7% improvement. Similarly, adding context improves accuracy by 2.3% when multi-scale features are not used, and by 2% when they are. These results confirm the effectiveness of both multi-scale and contextual features, independently and in combination. Tab. 2(c) investigates the impact of different context injection strategies. Results show that appending context to the full prompt (P) generally outperforms partial injection (V). The best accuracy (88.6% $\pm$ 2.7) is achieved when both tumor and normal context are used with full-prompt injection. Removing either context source results in a consistent performance drop, confirming the complementary value of tumor and normal contextual signals.

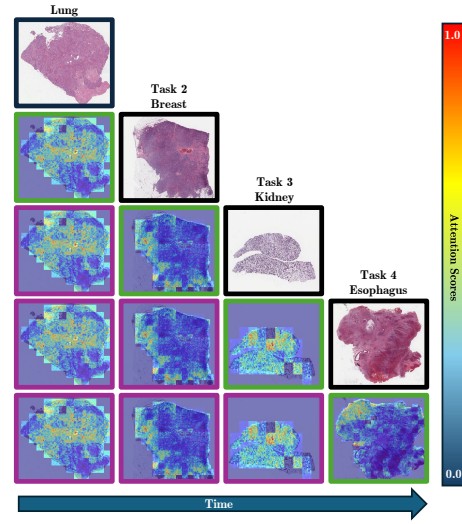

Figure 3: Patch-level attention maps over four consecutive tasks.

***Is localization stable over time?*** In Fig. 3 we present a qualitative analysis showing the model's ability to maintain consistent attention to relevant image regions across sequential tasks. Visualizations reveal stable localization for both current and past tasks (green and violet, respectively), highlighting the model's capacity for knowledge retention. This is especially important in medical contexts, where consistent and interpretable localization across resolutions and tasks enhances clinical trust and decision-making.

## 5 Conclusion

This work addresses key challenges in continual learning for WSI classification, including catastrophic forgetting, large-scale image analysis, multi-resolution processing, and privacy concerns. By integrating critical information into learnable prompts, CooMIL enhances classification performance and context awareness. Evaluations on four WSI datasets show improved accuracy and reduced forgetting. Despite these advantages, limitations remain. Although aligned with the existing literature, the benchmark tasks are relatively simple, and future work should incorporate more diverse imaging conditions and classification scenarios. While our approach involves incremental parameter growth, the overhead is minimal. Exploring prompt learning within the vision encoder also offers promising future directions.

## Acknowledgments and Disclosure of Funding

This work was supported by MUR under the National PRIN Project "AIDA: explAinable multImodal Deep learning for personAlized oncology".

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
