# OpenReview forum: "Context-guided Prompt Learning for Continual WSI Classification"
_MICCAI.org/2025/Workshop/COMPAYL — COMPAYL 2025_

### Official Review · Reviewer_Se9t · 2025-07-13
**Novel and relevant framework for continual learning on WSIs which is rehearsal-free**

**Rating:** 5
**Confidence:** 4

**Review:**

# Summary

The paper proposes CooMIL, a prompt‑based framework for continual classification of whole‑slide images (WSIs), which, importantly, is rehearsal-free. By injecting slide‑level context into learnable prompts and storing lightweight “prompt banks”, the method prevents catastrophic forgetting. The main motivation why rehearsal-free continual learning should be studied for WSIs is that of privacy.

On four TCGA cancer datasets CooMIL cuts forgetting to 3-5% and matches or exceeds rehearsal baselines, closing most of the gap to joint training. The paper is well-written with extensive experiments.

# Strengths

- The problem of continual learning for WSIs is well-motivated (already in the abstract). Rehearsal buffers (in prior existing approaches) raise privacy concerns in medical settings, so a rehearsal‑free alternative for gigapixel WSIs is valuable.
- CooMIL is the first continual learning framework for WSI classification that employ MIL and, importantly, is rehearsal free.
- CooMIL significantly outperforms existing approaches (increases accuracy by over 20% compared to ConSlide)
- The authors perform extensive and interesting ablation experiments (sec. 4.3) which strengthen their contributions.
- Code will be released and hyperparameters are disclosed.

# Weaknesses

- All four tasks derive from TCGA; no external validation cohorts (e.g., CPTAC) are used to test generalisability or domain shift.
- While CooMIL is rehearsal-free (avoids raw‑data rehearsal), the word bank stores task‑specific parameters that may leak private information; this risk is not discussed

# Comments

- It would be interesting if the authors provided the differences in parameter count between their method and the baselines.

# Conclusion

I recommend to accept this paper for the workshop, due to the well‑motivated and novel contribution which in my opinion outweighs the noted limitations.

---

### Official Review · Reviewer_dDRU · 2025-07-14
**Many missing aspects, including proper comparison, clinical motivation**

**Rating:** 2
**Confidence:** 4

**Review:**

Summary: The authors proposed an image-guided prompt learning approach for continual learning of Whole Slide Image Classification. The method is validated on TCGA dataset.

Strengths:
1. The technical idea is nice and intuitive.
2. The figures are well designed.

Weaknesses:
1. A major problem is the authors didn't acknowledge and compare with the related work on continual leaning in medical imaging. Even a Google Scholar search shows the first work on prompting a foundation model in medical continual learning setting is a MIDL24 article called Uncle SAM. I dug a bit deeper, and it seems like the author of that paper published a series of articles, none of which this paper compares to: https://scholar.google.de/citations?user=SujEnjMAAAAJ
2. I am also unsure about the author's claim on the privacy restrictions of the WSIs. As far as I know, HIPAA does not say, in general histopathology WSIs can contain patient specific information. I'd appreciate a deep discussion on this by the authors.
3. The equations are not very well thought out. For example, the equations in section 3.1 and 3.2 is not intuitive at all.
4. The continual learning experiment designed here is about changing organs. As such, what really is class incremental learning when the authors are performing slide level classification? The authors should properly define what they are trying to achieve.

---

### Official Review · Reviewer_GKvJ · 2025-07-14

**Rating:** 4
**Confidence:** 4

**Review:**

**Summary**

This paper introduces CooMIL, the first rehearsal-free continual learning framework for whole-slide image classification. It addresses the challenge of catastrophic forgetting in privacy-sensitive medical settings by avoiding data replay. Evaluated on four classification datasets from TCGA, CooMIL outperforms existing continual learning methods, including those that rely on memory buffers, and demonstrates strong performance in both accuracy and retention, while maintaining privacy compliance and offering greater computational efficiency.

**Strengths**

- the method is bit complex but presented with clear explanations, equations and figures
- all benchmarked methods rely on the same precomputed tile embeddings, ensuring fair comparison
- quick but effective ablation study to reveal the effectiveness of multi-scale and context-awareness
- nice visualisation of localisation preservation through attention maps

**Weaknesses**

- based on the preprocessing description, the CONCH vision encoder may not be used under optimal input size conditions: CONCH ViT-B expects inputs of 448×448, whereas the authors resize 512×512 patches to 224×224, which may degrade performance
- it would be preferable to report image resolution in microns per pixel (mpp) rather than magnification levels, as different scanners may yield different pixel spacing at the same magnification level, which can be confusing

**Detailed comments**

It is unclear why, during nearest neighbor search in the continual word bank, the authors used mean-pooled instance features rather than the MIL-pooled bag embedding, which should better encapsulates slide-level information.

While the authors emphasize that CooMIL is a rehearsal-free framework, it still stores centroids derived from whole-slide embeddings as part of the continual word bank. Although these are aggregated representations, they originate from potentially sensitive patient data. It would be helpful if the authors could clarify the privacy implications of retaining these centroids and whether they pose a risk under strict data governance settings.